# The Impact of Coronavirus Disease 2019 on Maternal and Fetal Wellbeing in New Mexico

**DOI:** 10.3390/diagnostics12112856

**Published:** 2022-11-18

**Authors:** Tiffany Emery, Kati Baillie, Orrin Myers, Hellen Ko, Jessie R. Maxwell

**Affiliations:** 1Department of Obstetrics and Gynecology, Jersey Shore University Medical Center, Neptune, NJ 07753, USA; 2Department of Pediatrics, Rainbow Babies and Children’s Hospital, Case Western Reserve University, Cleveland, OH 44106, USA; 3Department of Family and Community Medicine, University of New Mexico, Albuquerque, NM 87131, USA; 4Department of Pediatrics, University of New Mexico, Albuquerque, NM 87131, USA; 5Department of Neurosciences, University of New Mexico, Albuquerque, NM 87131, USA

**Keywords:** COVID-19, SARS-CoV-2, pregnancy, birth outcomes, newborn

## Abstract

Coronavirus disease 2019 (COVID-19) has been shown to affect the vasculature, including placental changes. Insults to the placenta, especially in the first and second trimester, can affect placental functionality with a resultant impact on fetal growth and wellbeing. Thus, we explored the relationship between antenatally acquired maternal COVID-19 infection and neonatal birth characteristics. A retrospective chart review was completed using the University of New Mexico electronic medical record system. ICD-10 codes were used to identify individuals that had a positive pregnancy test and positive COVID-19 screening test between 1 March 2020 to 24 March 2021. Chi-square and nonparametric Wilcoxon analyses were used, with *p* < 0.05 considered significant. A total of 487 dyad charts was analyzed, with 76 (16%) individuals identified as being COVID-19-positive (CovPos) during pregnancy. CovPos mothers were significantly more likely to deliver via a cesarean section compared to CovNeg mothers (33% vs. 20%, *p* < 0.01). There was a significant difference in gestational age at delivery, with infants born to CovPos individuals born at an earlier gestational age than those born to CovNeg individuals (37.6 vs. 38.5 weeks; *p* < 0.01). Our findings showed differences in maternal and infant characteristics following COVID-19 infection during pregnancy. Additional investigations are required to further delineate these relationships with a focus on potential long-term impacts on the neonate.

## 1. Introduction

On 20 January 2020, the first case of severe acute respiratory syndrome coronavirus-2 (SARS-CoV-2), also known as coronavirus disease 2019 (COVID-19), was confirmed in the United States. On 11 March 2020, the World Health Organization (WHO) declared COVID-19 a global pandemic. According to the Centers for Disease Control and Prevention (CDC) COVID Data Tracker, between 21 January 2020 and 15 June 2022, there was a total of 85,681,615 confirmed cases and 1,007,374 total deaths from COVID-19 in the United States [1]. Between 22 January 2020 and 13 June 2022, there was 217,210 total cases and 295 total deaths reported to occur during pregnancy within the United States [1]. Severe acute respiratory syndrome coronavirus-1 (SARS-CoV-1) and Middle East respiratory syndrome (MERS) are two illnesses caused by a coronavirus that preceded SARS-CoV-2. SARS-CoV-1 and MERS have been shown to affect maternal and neonatal morbidity and mortality [2,3,4,5,6]. Specifically, studies revealed that SARS-CoV-1 and MERS were associated with an increased rate in preterm births, pre-eclampsia, miscarriage, the preterm premature rupture of membranes, fetal growth restriction, cesarean section, and perinatal death [2,3,4,5,6]. Given that COVID-19 is a novel coronavirus, there are still many unknowns and concerns pertaining to COVID-19 in pregnancy, including maternal outcomes, neonatal outcomes, and the possibility of a vertical transmission. SARS-CoV-2 has also been shown to be more infectious and transmissible than SARS-CoV-1, adding to the concern for infection during pregnancy [7].

The transmission of COVID-19 occurs primarily through respiratory droplets. At a cellular level, a number of studies found that the co-expression of the ACE-2 receptor and the serine protease TMPRSS2 is required for SARS-CoV-2 to gain cytoplasmic entry and, therefore, infect the cell. Placental tissue has been shown to have a paucity of the ACE-2 receptor and TMPRSS2, arguing against the likelihood of a vertical transmission [7,8,9,10,11]. The vertical transmission of COVID-19 has been a controversial and inconclusive topic throughout the course of the pandemic. However, more recent research shows that vertical transmission may be possible, though rare [7,11,12,13,14,15,16,17].

The placenta is a vital organ of pregnancy that plays a key role in both maternal and fetal health. Since its emergence, COVID-19 has been shown to affect the vasculature and cause placental changes, such as microcalcifications, increased fibrin levels, and thrombi formation [18,19,20,21]. Well-known effects of COVID-19 on the placenta are vascular malperfusion and placental inflammation, which can lead to a placental insufficiency and fetal hypoxic–ischemic injury and have been hypothesized to be the mechanism for fetal death following maternal infection with COVID-19 [22]. A series of findings termed the “SARS-CoV-2 placentitis triad” has been described as consisting of intervillous/perivillous fibrin deposition, necrosis, and histiocytic intervillositis [22,23,24]. In addition, maternal viral infections during pregnancy can lead to profound immune activation and inflammation, which could have implications for the developing fetal brain and fetal neurodevelopmental outcomes [25].

COVID-19 has been shown to have adverse maternal and neonatal outcomes, including increased rates of preterm births, cesarean sections, intrauterine growth restrictions, low birthweights, hypertensive disorders of pregnancy, thrombotic events, fetal distress, APGAR scores < 7, NICU admissions, stillbirths, and maternal and neonatal deaths [4,5,19,26,27,28,29,30,31,32,33,34]. The increased rate of preterm births has been affected by increased maternal indications for delivery, such as worsening COVID-19 infection and increased rates of hypertensive disorders of pregnancy [32].

Additionally, some adverse outcomes may not manifest for months to years. Recent research has demonstrated preliminary findings of adverse neurodevelopmental outcomes, such as developmental disorders of language and speech as well as motor function, in children born to mothers who developed COVID-19 infection during their pregnancy that persisted after controlling for prematurity [35]. Another study found no differences at six months in infants who were exposed to COVID-19 in utero, but noted neurodevelopmental differences, such as lower gross and fine motor scores, in both the exposed and unexposed control groups compared to infants born before the COVID-19 pandemic. This study raises the question of whether the pandemic alone rather than COVID-19 infection during pregnancy may be responsible for neurodevelopmental outcomes [36].

In this study, we sought to explore the relationship, if any, between antenatally acquired maternal COVID-19 infection and fetal growth. We also evaluated the relationship between the timing of antenatal infections and neonatal growth parameters given the crucial period for placental remodeling and possibility for differential effects depending on the gestational age at the time of infection.

## 2. Materials and Methods

A retrospective chart review was completed using the University of New Mexico Hospital (UNMH) electronic medical record system for the time period of 1 March 2020 to 24 March 2021. International Classification of Diseases-10 (ICD-10) codes were used to identify mother and child dyad charts for the study, in which the maternal COVID-19 status was known. The ICD-10 codes used to identify COVID-19-positive individuals for review are shown in Table 1. Additionally, the ICD-10 codes used to identify pregnant women were Z33.1—pregnancy (single) (uterine)—and O30.9—pregnancy (multiple).

Table 1 shows ICD-10 codes used to identify study participants who were pregnant and COVID-19-positive.

Inclusion criteria consisted of a positive pregnancy test and positive COVID-19 screening test at UNMH between 1 March 2020 to 24 March 2021 and the existence of one live-born infant delivered at UNMH. Individuals with a positive pregnancy test and a negative or undocumented COVID-19 test without symptoms were included as a comparison group. One individual did not have a COVID-19 test and had no symptoms, and so was included in the comparison group. The remaining individuals all had a negative documented COVID-19 test if included in the comparison group. Charts were excluded if a live-born infant was not born at UNMH or if COVID-19 test results or symptoms were not identified (e.g., termination of pregnancy, discharge before delivery).

Variables identified from maternal charts included age, ethnicity, trimester of COVID-19 infection, maternal hospitalization needed for COVID-19 infection, maternal tobacco use, maternal medications, and maternal diagnoses or medical conditions. Variables identified from corresponding neonatal charts included sex, delivery mode (vaginal or cesarean section), gestational age, birthweight, birthweight category, fronto-occipital circumference (head circumference), APGAR scores at one and five minutes, and admission to routine newborn nursery versus higher-level admission to the neonatal intensive care unit or the intermediate care nursery. The birthweight categories included appropriate for gestational age (AGA), large for gestational age (LGA), symmetrically small for gestational age (sSGA), and asymmetrically small for gestational age (SGA). Standard definitions were used for birthweight categories: AGA 10–90th percentile, LGA > 90th percentile, sSGA < 10th percentile in weight and head circumference, SGA < 10th percentile in weight only.

Frequencies and percentages were calculated for categorical variables, and continuous variables were summarized using the mean, standard deviation, and median. Chi-square analyses were used for categorical data comparisons, and nonparametric Wilcoxon tests were used for continuous variables. Expected frequencies from chi-square analyses were examined to help interpret results. Logistic regression was completed for selected outcomes. A *p* value of < 0.05 was considered significant.

## 3. Results

Seventy-six mother and child pairs were identified for the CovPos group. A total of 411 women and children was included in the CovNeg group. One individual declined COVID-19 testing, but was asymptomatic and was, therefore, included in the comparison CovNeg group. The remainder of individuals in the CovNeg group (n = 410) had documented negative COVID-19 testing performed (See Figure 1 below).

From the 490 women and infant dyad charts initially identified for inclusion into our study, there was a total of 487 dyads that was included in the analyses. Three dyads were excluded, two for being duplicate records, and the other for the induction of a nonviable infant with a lethal chromosomal anomaly. Out of the 487 dyads, 76 (16%) women were found to have had COVID-19 during pregnancy and the remaining 411 (84%) women were included as a comparison group. Five individuals were positive during the first trimester, with the majority of individuals testing positive in the third trimester (n = 55; 72%). The individuals tested in the first trimester had testing completed due to the presence of concerning symptoms for infection. Antibody testing results were not included as data collection began prior to the approval of COVID-19 antibody testing by the United States Food and Drug Administration.

As seen in Table 2, the majority (72%) of women who were CovPos had a positive test during the third trimester of pregnancy and did not require hospitalization for COVID-19 (78%). However, 20% of individuals that were COVID-19-positive did require hospitalization due to symptoms related to the illness. There were no significant differences in the frequency of hypertensive disorders (pre-eclampsia, chronic hypertension, or gestational hypertension) or current, former, or never smokers between the two maternal groups. However, CovPos dyads were less likely to be non-Hispanic white or Asian, and more likely to be an American Indian/Alaska native (race/ethnicity *p* = 0.02) compared to CovNeg dyads. The incidence of C-sections (*p* = 0.01) and preterm births (*p* = 0.002) was also more common in CovPos individuals (Figure 2 and Figure 3). The indication for C-section delivery in CovPos individuals was directly related to COVID-19 infection in 24% of the cases (n = 6), all occurring during the third trimester.

For the infants, there were no significant differences in the sex of the child, one- and five-minute APGAR scores, frequency of small for gestational age or symmetrically small for gestational age infants, or admission to the newborn nursery versus the NICU or ICN, as shown in Table 3. Distributions of birthweight categories were not the same between the two groups (*p* = 0.01), with LGA infants more common in the CovPos (12%) than in the CovNeg (3%, *p* = 0.001) group (see Table 3).

Table 4 shows the birthweight and head circumferences of the infants with no significant differences between the two maternal groups. However, the mean gestational age was lower in CovPos infants (37.9 weeks) compared to CovNeg infants (38.7 weeks, *p* = 0.005; Table 4).

Additionally, we assessed associations between significant characteristics from Table 2 and Table 3 (race/ethnicity, delivery mode, preterm birth, LGA, and admission to higher-level care) with other maternal and infant characteristics (Table 5). The combined race/ethnicity category appeared to be associated with gestational hypertension (*p* = 0.004). The delivery mode was associated with tobacco use and pre-eclampsia (*p* < 0.001). Preterm delivery and LGA were independently associated with the trimester of infection in CovPos deliveries (*p* = 0.01); hypertensive disorders, such as pre-eclampsia and chronic hypertension, were also associated with preterm delivery (*p* < 0.001 and *p* = 0.02, respectively). APGAR scores also showed an association with where the infant was admitted after birth (*p* < 0.001).

## 4. Discussion

Coronavirus disease 2019 (COVID-19), also known as SARS-CoV-2, has been shown to have numerous effects on maternal and neonatal outcomes, including increased rates of adverse maternal and neonatal outcomes such as an increased rate of a preterm birth, cesarean section, intrauterine growth restriction, low birthweight, hypertensive disorders of pregnancy, thrombotic events, fetal distress, APGAR scores < 7, NICU admission, stillbirth, and maternal and neonatal death [4,5,19,26,27,28,29,30,31,32,33,34]. In the present study, we demonstrated a significant difference between the mode and timing of delivery between the two maternal groups. We found a higher rate of cesarean section deliveries in CovPos mothers compared to CovNeg mothers. Possible reasons for this difference include the presence of maternal comorbidities, obstetrical indications, severity of maternal respiratory disease, and maternal COVID-19-positive status. However, current evidence suggests that a COVID-19-positive status is not an indication for elective cesarean section delivery, and the decision should be determined solely based on obstetrical indications and disease severity [37,38,39]. There is also insufficient evidence supporting a decreased vertical transmission of SARS-CoV-2 based on the delivery mode [38,40]. Our findings were in line with numerous studies that also reported an increased rate of cesarean sections in individuals with COVID-19 in pregnancy compared to individuals without COVID-19 infection during pregnancy [26,37,41,42,43,44]. In addition, a higher severity of COVID-19 infection has been reported to be associated with increased rates of cesarean sections [26,44]. Twenty percent of individuals with COVID-19 infection required hospitalization due to symptoms, which may correlate with a higher severity of infection.

Our results also showed increased rates of preterm deliveries in the CovPos group. This was consistent with a number of studies that also reported higher rates of preterm births in individuals who acquired COVID-19 in pregnancy [29,42]. The inflammatory process plays a vital role in triggering the delivery of neonates via the release of proinflammatory cytokines [45,46]. Maternal viral infections during pregnancy can lead to profound immune activation and inflammation [25]. This established link between inflammation and delivery could be one possible reason behind the increased rate of preterm deliveries in CovPos moms. More importantly, increased preterm delivery rates for individuals with COVID-19 in pregnancy are also due to the higher levels of clinical interventions necessary, including cesarean section, due to maternal and/or fetal indications [47,48,49]. It is, however, important to note that preterm delivery was also associated with pre-eclampsia and chronic HTN, which could predispose to preterm births.

Our results did not indicate a significant difference between one- or five-minute APGAR scores or admission to the newborn nursery versus a higher-level nursery (i.e., newborn intensive care unit or intermediate care nursery) between populations. This was interesting, as other comparison studies did find significant differences in 5 min APGAR scores and NICU admission rates [34]. However, our study found a higher number of NICU admissions in babies born to CovPos (n = 65) vs. CovNeg (n = 20) mothers at a *p* value of 0.07, nearing statistical significance. This finding may be more reflective of our institution’s policies at the time rather than clinical differences between the infant groups. COVID-19-positive mothers had the option of not rooming in with their infant, which typically meant admission to the NICU or ICN. Repeat studies with a larger sample size and reason for admission to the NICU or ICN may find a significant difference.

Finally, our study results did not demonstrate a significant difference in the frequency of hypertensive disorders (pre-eclampsia, chronic hypertension, or gestational hypertension) between the two groups. However, numerous studies have shown a relationship between COVID-19 infection and increased risk for developing hypertensive disorders of pregnancy such as pre-eclampsia [43,50,51,52,53,54,55]. Possible explanations for the difference in results between our study and these studies may be due to the timing of infection during pregnancy, the severity of COVID-19 infection, and parity. A study by Papageorghiou et al. showed an association between COVID-19 infection in pregnancy and the development of pre-eclampsia, especially in nulliparous women [52]. The authors reported that the severity of infection did not seem to be a factor for the development of pre-eclampsia in their study [52]. However, in a study by Metz et al., the severity of COVID-19 infection appeared to have a role in the development of hypertensive disorders of pregnancy [53]. Severe COVID-19 infection was associated with an increased risk of hypertensive disorders of pregnancy compared to individuals who were asymptomatic. In their study, there was no association between adverse outcomes and mild to moderate COVID-19 illness [53]. Finally, as discussed in the Introduction, crucial placental remodeling occurs throughout the first trimester and at the beginning of the second trimester. Therefore, it was hypothesized that the timing of COVID-19 infection may play a role in the development of hypertensive disorders of pregnancy. The majority (72%) of our study participants who were COVID-19-positive during pregnancy tested positive during the third trimester, well after crucial placental remodeling was complete. This could be one reason why we did not find a significant difference in the frequency of hypertensive disorders of pregnancy between the control and experimental groups. Similarly, as it is thought that hypertensive disorders can affect the growth of the fetus and there was no significant difference in the incidence of hypertensive disorders between our two study groups, it was not surprising that the incidence of SGA or sSGA infants did not differ significantly between the groups. However, it did not explain the finding of more LGA infants in the CovPos group.

Our results also indicated that there was a significant disparity in race/ethnicity and COVID-19-positive status. We demonstrated that women with a COVID-19 infection during pregnancy were more likely to be Native American than non-Hispanic white women. This correlates with CDC data showing a disparity among races in infection rates and severity worldwide. Black, Hispanic/Latino, and Native Americans are more likely to contract COVID-19 and have higher rates of COVID-19-related hospitalizations and deaths [56]. Furthermore, a systematic review consisting of 54 studies concluded that African American/Black and Hispanic populations experienced a disproportionate rate of COVID-19 infection and COVID-19-related mortality. Regarding American Indian/Alaska Native populations, the authors acknowledge that more data are needed to draw conclusions regarding disparities in COVID-19 infection rates and morbidity/mortality. However, the study also referenced data from the National Center for Health and Statistics, which show that American Indian/Alaska Native populations experience 1.9% excess deaths, and data from the CDC’s COVID-NET, which show that American Indian/Alaska Native individuals are four times more likely to be hospitalized due to COVID-19 infection compared to non-Hispanic white individuals [57]. A number of factors were hypothesized to play a role in this disparity, such as decreased access to healthcare, poverty, multigenerational homes, differential rates of low-paying jobs, as well as unemployment, increased rates of pre-existing co-morbidities, and racism. Race/ethnic disparities may stem from socioeconomic inequities; however, these disparities have been found to persist when accounting for socioeconomic factors. Therefore, we cannot ignore the well-studied effects of systemic/structural racism on health outcomes and as a mitigating factor of health disparities [57,58,59,60]. Given our study population, our study contributes to the limited data on COVID-19 in pregnancy in the Native American population, with most Native American patients in our population identifying as Navajo.

The results of this study have limitations. The first being that the study was conducted at a single academic institution and may not be generalizable to other settings. Future multicenter studies may be needed to demonstrate external validity. Further limitations of this study included the study population being restricted to only women with a documented COVID-19 test in our hospital system. In addition, while our study had a large number of participants enrolled overall, only 16% (76/488) of study participants that were included comprised the experimental group. Additionally, the study was completed over a period in which vaccines became available. Data on vaccination rates and potential impacts on the health of the pregnant individual are being reviewed for this cohort, but are beyond the scope of the current study. Finally, the majority of study participants had a positive COVID-19 test during the third trimester of pregnancy, so our results may not be as generalizable to individuals who acquire a COVID-19 infection in the first and second trimesters. Further research is needed to evaluate the differences in maternal and neonatal outcomes based on the timing of COVID-19 infection in pregnancy.

## 5. Conclusions

In conclusion, our study demonstrated several differences in maternal and neonatal outcomes in individuals who acquired COVID-19 infection during pregnancy compared to individuals who did not acquire COVID-19 infection during pregnancy. Compared to CovNeg mothers, CovPos mothers had statistically significant increases in the rates of cesarean sections and preterm deliveries. Our results also showed statistically significant differences in the growth parameters of the infants. Finally, our study suggested a disparity between race/ethnicity and COVID-19 status, with a lower incidence of COVID-19 infection in non-Hispanic whites and Asian women and a higher incidence of COVID-19 in American Indian/Alaskan Natives, though a larger study is necessary to provide support.

## Figures and Tables

**Figure 1 diagnostics-12-02856-f001:**
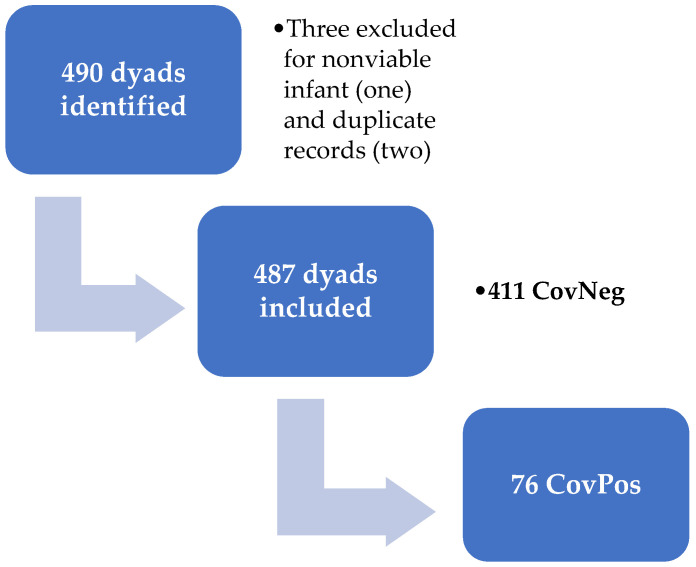
Identification of participants for inclusion.

**Figure 2 diagnostics-12-02856-f002:**
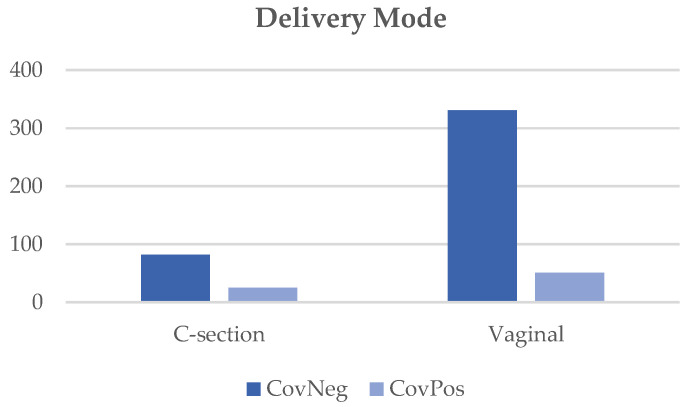
Mode of delivery between CovNeg and CovPos groups.

**Figure 3 diagnostics-12-02856-f003:**
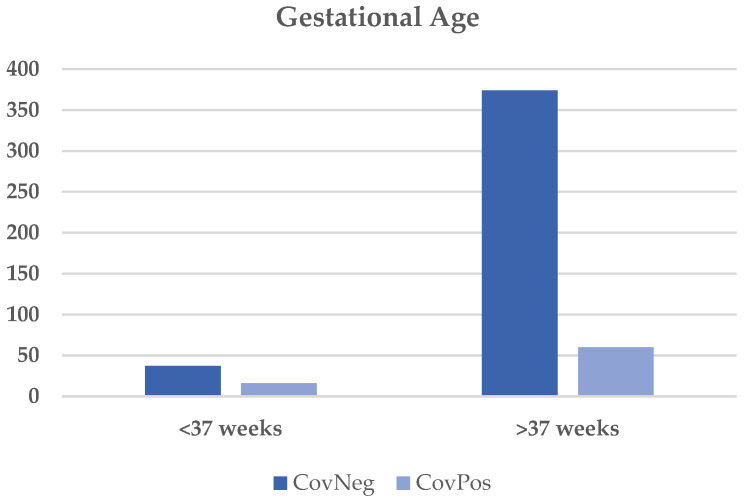
Gestational age at delivery between CovNeg and CovPos groups.

**Table 1 diagnostics-12-02856-t001:** ICD-10 Pregnancy and COVID-19 codes used to identify charts for review.

ICD 10 Codes	
B34.2	Coronavirus infection, unspecified
B97.2	Coronavirus as the cause of diseases classified elsewhere
B97.29	Other coronavirus as the cause of diseases classified elsewhere
J12.81	Pneumonia due to SARS-associated coronavirus
Z20.822	Contact with and (suspected) exposure to COVID-19
Z11.52	Encounter for screening for COVID-19
Z86.16	Personal history of COVID-19
M35.81	Multisystem inflammatory syndrome
J12.82	Pneumonia due to coronavirus disease 2019
Z33.1	Pregnancy (single) (uterine)
030.9	Pregnancy (multiple)

**Table 2 diagnostics-12-02856-t002:** Maternal demographics of CovNeg and CovPos groups.

	Maternal COVID-19 Infection During Pregnancy			
Demographics	Overall (n = 487)	No (n = 411)	Yes (n = 76)	*p*-Value
	**n (%)**	**n (%)**	**n (%)**	
Trimester of Infection				
1			5 (7)	
2			6 (21)	
3			55 (72)	
Race/ethnicity				0.02
American Indian/Alaska Native	66 (14)	50 (12)	16 (21)	
Hispanic	263 (54)	218 (53)	45 (59)	
Non-Hispanic white	82 (17)	77 (19)	5 (7)	
Asian	14 (3)	14 (3)	0 (0)	
Black/AA *	18 (4)	16 (4)	2 (3)	
Other/no answer	44 (9)	36 (9)	8 (11)	
Sex of child				0.77
Female	236 (48)	198 (48)	38 (50)	
Male	251 (52)	213 (52)	38 (50)	
Delivery mode				0.01
C-section	107 (22)	82 (20)	25 (33)	
Vaginal	380 (78)	331 (80)	51 (67)	
Hospitalization for COVID?				
n/a		411 (100)	-	
No		-	59 (78)	
Yes		-	15 (20)	
Unknown		-	2 (2)	
Pre-eclampsia				0.97
No	430 (88)	363 (88)	67 (88)	
Yes	57 (12)	48 (12)	9 (12)	
Chronic HTN +				0.49
No	457 (94)	387 (94)	70 (92)	
Yes	30 (6)	24 (6)	6 (8)	
Gestational HTN				0.14
No	439 (90)	367 (89)	72 (95)	
Yes	48 (10)	44 (11)	4 (5)	

* **AA**, African American; ^+^
**HTN**, hypertension.

**Table 3 diagnostics-12-02856-t003:** Infant demographics between CovPos and CovNeg groups.

	Maternal COVID-19 Infection During Pregnancy			
	Overall (n = 487)	No (n = 411)	Yes (n = 76)	*p*-Value
	**n (%)**	**n (%)**	**n (%)**	
Gestational Age				0.002
<37 weeks	53 (11)	37 (9)	16 (21)	
≥37 weeks	434 (89)	374 (91)	60 (79)	
Birthweight Category **				
AGA				0.01
LGA	408 (84)	348 (85)	60 (79)	
SGA	23 (5)	14 (3)	9 (12)	
sSGA	30 (6)	25 (6)	5 (7)	
	26 (5)	24 (6)	2 (3)	
Any SGA				
Yes				
No	56 (11)	49 (12)	7 (9)	0.50
	431 (89)	362 (88)	69 (91)	
Any LGA				
Yes	23 (5)	14 (3)	9 (12)	0.001
No	464 (95)	397 (97)	67 (88)	
Admit ^++^				0.07
NBN	385 (79)	330 (80)	55 (72)	
NICU	85 (17)	65 (16)	20 (26)	
ICN	16 (3)	15 (4)	1 (1)	
APGAR (1 min)				0.37
Abnormal (0–3)	23 (5)	17 (4)	6 (8)	
Mod Abnormal (4–6) ^	45 (9)	38 (9)	7 (9)	
Normal (7–10)	417 (85)	354 (87)	63 (83)	
APGAR (5 min)				0.50
Abnormal (0–3)	7 (1)	6 (1)	1 (1)	
Mod Abnormal (4–6) ^	16 (3)	12 (3)	4 (5)	
Normal (7–10)	462 (95)	391 (96)	71 (93)	

** **AGA**, appropriate for gestational age (growth parameters, i.e., birthweight and head circumference, are in the 10–90th percentile); **LGA**, large for gestational age (growth parameters > 90th percentile); **SGA**, small for gestational age (weight < 10th percentile); **sSGA**, symmetrically small for gestational age (birthweight and head circumference < 10th percentile); ^++^
**NBN**, newborn nursery; **NICU**, neonatal intensive care unit; **ICN**, intermediate care nursery; ^ **Mod abnormal**, moderately abnormal.

**Table 4 diagnostics-12-02856-t004:** Infant growth parameters and maternal COVID-19 status.

	Maternal COVID-19 Infection								*p*-Value
	No				Yes				
	n	Mean	SD	Median	n	Mean	SD	Median	
**Birthweight (g)**	409	3144	546	3190	76	3064	769	3168	0.37
**Head Circumference**	410	34	2	34	76	34	2	34	0.28
**Gestational Age (Weeks)**	411	38.7	2.3	38.9	76	37.9	2.5	38.9	0.005

**Table 5 diagnostics-12-02856-t005:** Association between selected factors and other maternal and infant characteristics.

Factors from Table 2	Associations	*p*-Value
Race/Ethnicity	Gestational HTN	0.004
Delivery Mode	Tobacco use	<0.001
	Pre-eclampsia	<0.001
GA * < 37 Weeks	Trimester of infection	0.01
	Pre-eclampsia	<0.001
	Chronic HTN	0.02
Any LGA	Trimester of infection	0.01
Admit	APGAR (1 Min)	<0.001
	APGAR (5 Min)	<0.001

* GA, gestational age.

## Data Availability

The data presented in this study are available on request from the corresponding author. The data are not publicly available due to the inclusion of protected health information.

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
