# Peer review of "The Impact of Coronavirus Disease 2019 on Maternal and Fetal Wellbeing in New Mexico"

_diagnostics, 2022, doi:10.3390/diagnostics12112856_

Round 1
Reviewer 1 Report
The topic of the article is interesting and the basic statistics has been well conducted. However there are some key-points to clarify.
1- We do not have any anamnestic data, necessary at least in the 76 cases. Is there any pregnant woman who has contracted Covid-19 before pregnancy? Furthermore, it is not clear whether women who gave birth early had other pathological conditions predisposing to preterm birth.
2- It is not clear how many women had a simple positivity or a disease linked to Covid-19.
3 - The same applies to women undergoing an elective caesarean section; had they had other complicated natural births or caesarean sections? Furthermore, the authors state: “We found a higher rate of cesarean section delivery in CovPos mothers 206 compared to CovNeg mothers. Possible reasons for this difference include presence of 207 maternal comorbidities, obstetrical indications, severity of maternal respiratory disease, and maternal COVID positive status ". For the selected sample, authors should be more accurate and better establish:
a) the causes that led to performing the caesarean section;
b) if the aforementioned are related to Covid-19 infection, specifying in this case the period of pregnancy in which the virus was contracted.
4- Important limitation of the study: there is no anamnestic data on the Covid-19 vaccination of the women recruited in the study. Literature claims that pregnant and breastfeeding women are at an increased risk of severe illness and Covid-19 vaccination is the most effective way for people who are pregnant to protect themselves and their babies. Authors should include some specific data on this topic.
Author Response
1- We do not have any anamnestic data, necessary at least in the 76 cases. Is there any pregnant woman who has contracted Covid-19 before pregnancy? Furthermore, it is not clear whether women who gave birth early had other pathological conditions predisposing to preterm birth.
-Thank you to reviewer #1 for the points raised. We have clarified that antibody testing is not included because FDA did not approve the first antibody test until after the start of the data collection period, and thus would not be complete information. We have clarified in lines 179-184 the above point, as well as clarifying that the 5 individuals that were positive in the first trimester were testing due to the presence of symptoms. We have added to the results text to stress the results of Table 5, which show that preterm delivery was associated with the trimester of COVID-19 infection independently, as well as pre-eclampsia, and chronic hypertension.
2- It is not clear how many women had a simple positivity or a disease linked to Covid-19.
-Of the positive results, 20% of the individuals required hospitalization due to symptoms of Covid-19. This has also been clarified in the results and the discussion section (lines 187-188 and 263-265).
3 - The same applies to women undergoing an elective caesarean section; had they had other complicated natural births or caesarean sections? Furthermore, the authors state: “We found a higher rate of cesarean section delivery in CovPos mothers 206 compared to CovNeg mothers. Possible reasons for this difference include presence of 207 maternal comorbidities, obstetrical indications, severity of maternal respiratory disease, and maternal COVID positive status ". For the selected sample, authors should be more accurate and better establish:
- the causes that led to performing the caesarean section;
- We have clarified that the indication for the cesarean section delivery in COVID-19 positive individuals was directly related to COVID infection in 24% of cases, all of which occurred in the third trimester. This has been added to lines 194-196.
- if the aforementioned are related to Covid-19 infection, specifying in this case the period of pregnancy in which the virus was contracted.
- We have clarified that the indication for the cesarean section in COVID-19 positive individuals was 24%, all of which occurred in the third trimester. This has been added to lines 194-196.
4- Important limitation of the study: there is no anamnestic data on the Covid-19 vaccination of the women recruited in the study. Literature claims that pregnant and breastfeeding women are at an increased risk of severe illness and Covid-19 vaccination is the most effective way for people who are pregnant to protect themselves and their babies. Authors should include some specific data on this topic.
- Vaccination is an interesting area of discussion, and we are currently reviewing that information for this cohort to further investigate this point. However, as vaccines were available for only a part of this cohort, that data is not being included in this manuscript, but will be reviewed for further work.
Reviewer 2 Report
In this manuscript authors aimed at evaluating the relationship between timing of antenatal COVID-19 infections and neonatal growth parameters given the crucial period for placental remodeling and possibility for differential effects depending on gestational age at the time of infection. This is an interesting topic as the impact of the disease on maternal and fetal outcomes has been questioned since the pandemic began.
Comment:
The introduction is to long and I believe should be shortened. There are several repeated points.
Methods:
The study suffers major flaws in its design or analysis that the conclusions can not be supported by the presented data. The way the comparison -control group was identified is the main problem of the study. Authors reported that “Individuals with a positive pregnancy test and undocumented COVID test without symptoms were included as a comparison group.”
Author Response
The introduction is to long and I believe should be shortened. There are several repeated points.
- We appreciate the comments provided by reviewer #2. The introduction has been shortened to minimize any repeated points.
Methods:
The study suffers major flaws in its design or analysis that the conclusions can not be supported by the presented data. The way the comparison -control group was identified is the main problem of the study. Authors reported that “Individuals with a positive pregnancy test and undocumented COVID test without symptoms were included as a comparison group.”
- We have clarified that only 1 individual did not have any covid testing completed (but had no symptoms). The remainder of individuals that were included in the comparison group were truly negative based on COVID testing. This information is found on lines 140-143 and lines 170-171.
Round 2
Reviewer 1 Report
While some small research limitations remain, the authors have provided adequate answers and made the required changes.
Reviewer 2 Report
In this version of manuscript the authors made great effort to improve the article.